# Comparison of medical students' perceptions of patient safety: Focusing on simulation training using a high-fidelity simulator

**Ji Eun Lee** [1], **Ji Hye Yu**[1,2], **Su Kyung Lee**[3], **Jang Hoon Lee**[1,2,4], **Hyun Joo Jung** [4] *

1 Office of Medical Education, Ajou University School of Medicine, Suwon, South Korea, 2 Department of Medical Education, Ajou University School of Medicine, Suwon, South Korea, 3 Ajou Center for Clinical Excellence, Ajou University School of Medicine, Suwon, South Korea, 4 Department of Pediatrics, Ajou University School of Medicine, Suwon, South Korea

* free1109@ajou.ac.kr

## Abstract

Patient safety education is necessary for the provision of high-quality medical services. A significant aspect of patient safety education is simulation training, which allows medical students to experience realistic clinical environments. This study aimed to verify the effectiveness of patient safety education using simulation training. We retrospectively analyzed the results of a 30-question questionnaire survey on the perceptions of patient safety before and after simulation training, which was completed by 40 medical students who participated in clinical practice between June and December 2021. A paired t-test was performed by calculating the mean and standard deviation for each item. We found that students' overall perceptions of patient safety improved after training. Specifically, after simulation training, attitudes toward patient safety were maintained at the same level as before training, while students' self-efficacy of patient safety increased. Simulation training is effective in improving students' perceptions of patient safety, and increasing students' confidence can improve their clinical performance. To maintain this effect, repeated learning is required, and theoretical classes and simulation training should be used appropriately for patient safety education in the future.

## Introduction

Rapid social development and economic growth have resulted in increased demand and expectations for healthcare services. However, with increasing access to healthcare services, the incidence of medical accidents is also increasing. Every year, there are millions of instances of disability, injury, or death due to unsafe medical practices [1], and this has resulted in increased awareness of the importance of patient safety and heightened interest in improving the quality of healthcare services.

Medical accidents cause physical, mental, and material losses for patients, and accidents related to patient safety can be fatal [2]. Therefore, medical institutions and their members should strive to provide quality healthcare services. According to the Institute of Medicine's

**Data Availability Statement:** "Due to confidentiality agreements, supporting data can only be made available to bona fide researchers subject to a non-

disclosure agreement. Details of the data and how to request access are available from the corresponding author of this study and the data access committee. Data are available from the following Data Access Committee members: Ji Eun Lee via phone (82-10-3929-7011) or email (jieun920733@gmail.com), Ji Hye Yu via phone (82-31-219-4092) or email (blue8106@aumc.ac.kr), Su Kyung Lee via phone (82-31-219-4463) or email (esukyung71@gmail.com), Jang Hoon Lee via phone (82-31-219-5167) or email (neopedlee@aumc.ac.kr), and Hyun Joo Jung via phone (82-31-219-5160) or email (free1109@ajou.ac.kr), for researchers who meet the criteria for access to confidential data. Data are also available from Ajou University IRB Ethics Committee (https://eirb.ajoumc.or.kr/) via phone (82-31-219-5569) or email (aumc_qa@aumc.ac.kr), for researchers who meet the criteria for access to confidential data."

**Funding:** This research was supported by a grant of the Korea Health Technology R&D Project through the Korea Health Industry Development Institute (KHIDI), funded by the Ministry of Health & Welfare, Republic of Korea (grant number: HG22C003500). The funders had no role in study design, data collection and analysis, decision to publish, or preparation of the manuscript.

**Competing interests:** The authors have declared that no competing interests exist.

1999 "To Err is Human" report, medical errors are more common than we think, and many are preventable [3]. However, ensuring maximum patient safety in healthcare services is often difficult owing to changes in and increased complexity of healthcare processes, information overload for patients or healthcare providers, increased patient expectations for perfect treatment outcomes, and higher severity of patient conditions and vulnerability [4]. To ensure safe healthcare services, medical accidents must be prevented, and when such accidents do occur, the causes and consequences must be identified to prevent recurrence [5]. In addition, a safe medical system should be developed to effectively prevent and reduce errors and avoid blaming individuals in the event that they do occur [3]. To improve patient safety, processes and systems must be implemented to account for inevitable human errors.

Patient safety education is also becoming increasingly important in medical education. The Institute of Medicine emphasized the need for continuous patient safety education in medical education courses [6]. In line with this suggestion, the World Health Organization announced a patient safety curriculum for medical schools in 2009 [7], and efforts are being made globally to implement patient safety education in medical schools' basic medical curriculum. In Korea, some individual medical educators, not organizations such as schools, are still recognizing the importance of patient safety education and making efforts [8]. Therefore, since the timing and methods of education vary, it is necessary to consider when and how to provide education more effectively. In Korea, patient safety education in medical schools is mostly provided before clinical practice [8]. However, during the clinical practice period, students will meet patients in person, and medical errors may occur during this period. Therefore, patient safety education should be continuously provided during clinical practice.

Simulation training aims to provide learners with practice opportunities, enabling them to learn from mistakes in a safe environment and acquire mastery by achieving predetermined learning objectives [9]. Additionally, it allows for objective evaluation of learners' competency through standardized learning content, as well as repeated learning [10]. Therefore, it can be considered an effective educational method for improving students' competency in medical education, as it allows clinical situations to be designed around various learning goals. These advantages of simulation training can also be applied to patient safety education. In Korea, several previous studies have already verified the effectiveness of simulation training programs on patient safety accidents, such as blood transfusion errors and fall management, for prospective medical personnel [11, 12].

The need for all medical personnel to collaboratively strive for patient safety has been highlighted as a measure to prevent repeated medical accidents. Therefore, patient safety simulation training is essential for prospective medical personnel. In Korea, the effectiveness of simulation training has been verified in several studies that developed such training programs for prospective medical personnel, especially nursing students [11–13]. However, simulation training research on medical students is insufficient. Since 2006, many medical schools in Korea have been operating simulation centers [14]. Therefore, they also need to provide simulation training on patient safety for medical students. Thus, we aim to confirm the effectiveness of simulation training in patient safety education by analyzing medical students' perceptions of patient safety before and after simulation training.

## Materials and methods

### Design and setting

We conducted a survey with 40 students (28 men, 12 women) to compare their perceptions of patient safety. The students were juniors taking a clinical practice course on pediatrics and adolescence at a college of medicine in the academic year 2021. They had had three simulation

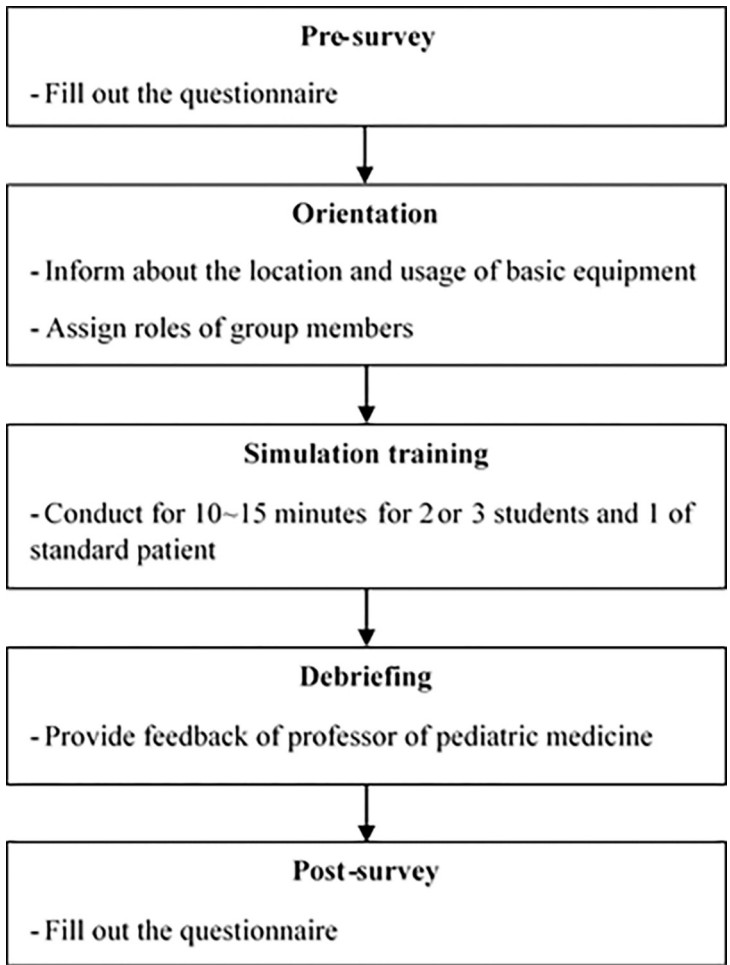

**Fig 1. Flow chart of the study.** A total of 40 students, engaged in clinical practice in pediatrics, underwent simulation training and completed a questionnaire both before and after the training.

training experiences in previous clinical practice. They participated in the training from June to December 2021, and we retrospectively analyzed the questionnaire data collected during the regular curriculum.

As part of the assessment, students filled out a pre-survey questionnaire on their perceptions of patient safety. Next, an orientation was conducted to inform students about the location and usage of basic equipment through pre-briefing. Then, students were divided into pairs, and each pair was assigned certain roles to facilitate task sharing. The training, supervised by a pediatric medicine professor, lasted 10–15 minutes. Following the training, debriefing sessions provided feedback on students' learning goals, focusing on their knowledge, skills, and attitudes. Finally, a post-training survey was conducted, employing the same questions as those used in the pre-survey questionnaire (Fig 1).

## Ethical considerations

This study was conducted with the approval of the concerned institutional review board (No. AJIRB-SBR-SUR-21-624). Informed consent was not required because the study involved a retrospective analysis based on previously collected data.

## Simulation program setting

In the four-week clinical practice course on pediatrics, students were divided into five groups of eight students each. Each group was further subdivided into pairs during the simulation training, which was conducted four times. The training duration for each pair was approximately 10 min on average and did not exceed 15 min. Since falls are one of the most common patient safety-related accidents among pediatric patients, pediatric falls were selected as the subject of the simulation program scenario in this study. A high-fidelity pediatric simulator (SimBaby; Laerdal Medical, Stavanger, Norway) was used to allow students to fully focus on the situation. Vital signs were displayed on the monitor, and test results suitable for the scenario were provided. For a more realistic scenario, one standardized patient playing the role of a patient's caregiver was recruited and trained in advance to perform the same role during each situation. The simulation training was designed to train students to (1) determine the severity of an adverse event through history-taking and physical examination; (2) communicate appropriately with patients and caregivers; and (3) understand the importance of reporting, analysis, and management of adverse events. The simulation training program was organized into four stages according to the learning objectives, as shown in Fig 2.

## Questionnaire

We used a questionnaire on the perceptions of patient safety. The questions were the same for both the pre- and post-survey questionnaires. However, the post-survey questionnaire included an additional question on students' feelings. There were 30 questions in all: 24 questions on the participants' attitudes toward overall safety, including falls, and six questions on self-efficacy related to procedures and methods for coping with falls. All items were answered on a 5-point Likert scale (1 = not at all; 5 = strongly agree).

To determine students' attitudes toward patient safety, we used an evaluation tool developed by Madigosky et al. [15], which determines the changes in medical students' knowledge, skills, and attitudes regarding patient safety and medical errors. Further, Kim and Seo's questionnaire was used to measure students' knowledge and attitude toward falls [16]. The questionnaire was modified and supplemented to meet the purpose of this study. In addition, Park and Park's questionnaire was used as a measure of self-efficacy for patient safety-related performance; it included six items related to coping with medical errors, falls, and safety issues [17]. A team of experts, including one medical education department professor, two professors from the department of pediatrics, and one nurse, corrected and supplemented the final questionnaire with any necessary explanations. The Cronbach alpha values of the 30 questions in the final completed questionnaire were 0.781 for the pre-survey and 0.816 for the post-survey.

## Statistical analysis

SPSS Statistics 25.0 (IBM Corp., Armonk, NY, USA) was used to analyze the data. Means and standard deviations were calculated for each item to determine students' perceptions of patient safety, and a paired $t$-test was performed to compare the pre- and post-training results.

## Results

Table 1 presents the results of students' perceptions of patient safety before and after the simulation training program. In particular, the post-training results demonstrated an improvement in students' perceptions of patient safety, showing a statistically significant difference.

Further, there was no statistically significant difference in students' attitudes toward patient safety after training, but the average scores above neutral were maintained. The results also

---

**Stage 1**

*"Found a crying baby under the bed at the same time as parents."*

- Learning objectives : Students can recognize of occurrence of medical errors.

**Stage 2**

*"Conduct history taking and physical examination."*

- Learning objectives : Students can judge the severity of a child's fall.

**Stage 3**

*"Explain the necessity of testing to the baby's parents and conduct the test."*

- Learning objectives : Students can honestly communicate with patients and their guardians about problems that have occurred.

**Stage 4**

*"Check and review the results of the examination performed and explain it to the baby's parents."*

- Learning objectives :

  Students can report the occurrence of an incident to their boss in an appropriate way.

  Students can understand the need for case analysis and follow-up management.

**Fig 2. Stages of the simulation program and learning objectives by stage.** The students' simulation training was conducted according to the steps shown here. It also aimed to achieve the learning objectives corresponding to each stage.

demonstrated a statistically significant level of positive change among students regarding the provision of time for patient safety education at school.

Additionally, the post-training results demonstrated a statistically significant increase in the degree of interest in inpatient falls compared to the pre-training results. The perception of the

**Table 1. Comparative analysis of students' perceptions of patient safety before and after simulation training.**

| Sub-variables | Questions | Mean ± SD | | t-value | p-value |
|---|---|---|---|---|---|
| | | Pre-test | Post-test | | |
| Attitude | 1. Physicians should not tolerate uncertainty in patient care. | 3.73 ± 0.72 | 3.68 ± 0.89 | 0.388 | 0.700 |
| | 2. Only physicians can determine the causes of a medical error. | 3.33 ± 1.05 | 3.30 ± 1.22 | 0.138 | 0.891 |
| | 3. Learning how to improve patient safety is an appropriate use of time in medical school. | 4.25 ± 0.63 | 4.55 ± 0.60 | −2.762 | 0.009* |
| | 4. Physicians routinely share information about medical errors and what caused them. | 3.85 ± 0.70 | 3.85 ± 0.74 | 0.000 | 1.000 |
| | 5. Physicians routinely report medical errors. | 3.75 ± 0.90 | 3.73 ± 0.82 | 0.206 | 0.838 |
| | 6. The culture of medicine makes it easy for providers to deal constructively with errors. | 2.83 ± 0.84 | 2.68 ± 0.89 | 1.183 | 0.244 |
| | 7. Physicians should routinely spend part of their professional time working to improve patient care. | 3.98 ± 0.62 | 4.10 ± 0.59 | −1.706 | 0.096 |
| | 8. There is a gap between what we know as "best care" and what we provide on a day-to-day basis. | 4.18 ± 0.78 | 4.25 ± 0.63 | −0.829 | 0.412 |
| | 9. If there is no harm to a patient, there is no need to address an error. | 3.76 ± 0.80 | 3.90 ± 0.74 | −1.152 | 0.256 |
| | 10. In my clinical experiences so far, faculty and staff have communicated to me that patient safety is a high priority. | 3.70 ± 0.85 | 3.93 ± 0.76 | −1.503 | 0.141 |
| | 11. If I saw a medical error, I would keep it to myself. | 4.08 ± 0.66 | 4.15 ± 0.62 | −0.771 | 0.446 |
| | 12. Reporting systems do little to reduce future errors. | 4.15 ± 0.66 | 4.28 ± 0.68 | −1.533 | 0.133 |
| | 13. Physicians should be the healthcare professionals who report errors to an affected patient and their family. | 3.90 ± 0.59 | 4.05 ± 0.75 | −1.356 | 0.183 |
| | 14. Competent physicians do not make medical errors that lead to patient harm. | 3.08 ± 1.02 | 3.15 ± 1.29 | −0.476 | 0.637 |
| | 15. Effective responses to errors focus primarily on the healthcare professional involved. | 3.18 ± 0.75 | 3.18 ± 0.98 | 0.000 | 1.000 |
| | 16. Making errors in medicine is inevitable. | 2.23 ± 0.73 | 2.25 ± 0.81 | −0.190 | 0.850 |
| | 17. I think I should respond to patients immediately if they ask for help when they move. | 3.53 ± 0.82 | 3.43 ± 0.87 | 0.752 | 0.457 |
| | 18. I think I should assess the danger of falling related to patients when they are hospitalized. | 4.35 ± 0.58 | 4.40 ± 0.59 | −0.495 | 0.623 |
| | 19. I think patients do not sustain much physical damage when they fall. | 4.23 ± 0.80 | 4.23 ± 0.77 | 0.000 | 1.000 |
| | 20. I am concerned about inpatients' accidental falls. | 3.50 ± 0.85 | 3.98 ± 0.77 | −2.829 | 0.007* |
| | 21. I will feel guilty if a patient falls. | 3.38 ± 0.87 | 3.28 ± 1.04 | 0.703 | 0.486 |
| | 22. I think falling in the hospital is an important responsibility of healthcare providers. | 3.68 ± 0.76 | 3.40 ± 0.98 | 1.921 | 0.062 |
| | 23. I think there is enough current fall prevention education for the patients when they are hospitalized. | 3.33 ± 0.89 | 3.43 ± 0.96 | −0.662 | 0.512 |
| | 24. I think falling is caused by the patient's condition. | 3.20 ± 0.85 | 3.53 ± 0.91 | −2.177 | 0.036* |
| | Average | 3.63 ± 0.28 | 3.69 ± 0.32 | 1.75 | 0.087 |
| Self-efficacy | 25. Be sure to lock beds and wheelchairs when transferring a client from a bed to a wheelchair or back to bed. | 2.55 ± 0.96 | 3.08 ± 1.02 | −3.667 | 0.001* |
| | 26. Use side rails appropriately and explain the importance of appropriate use of side rails. | 2.90 ± 1.10 | 3.95 ± 1.01 | −6.565 | 0.000* |
| | 27. Support and advise a peer who must decide how to respond to an error. | 3.60 ± 0.71 | 3.85 ± 0.74 | −1.883 | 0.067 |
| | 28. Disclose an error to a healthcare professional. | 3.63 ± 0.59 | 3.85 ± 0.77 | −1.778 | 0.083 |
| | 29. Analyze a case to find the causes of an error. | 3.68 ± 0.80 | 3.85 ± 0.80 | −1.361 | 0.181 |
| | 30. Accurately complete an incident report. | 3.25 ± 0.81 | 3.63 ± 0.90 | −2.831 | 0.007* |
| | Average | 3.27 ± 0.54 | 3.70 ± 0.66 | 5.61 | 0.000* |
| Average | | 3.56 ± 0.30 | 3.70 ± 0.34 | 3.63 | 0.001* |

severity of a patient's physical injury owing to a fall and the perception of the occurrence of a fall due to a patient's physical condition were above neutral.

Finally, the post-training results demonstrated a statistically significant increase in students' self-efficacy related to patient safety. In particular, self-efficacy in writing patient safety event reports increased after the training. The scores of the importance and proper use of the bed, wheelchair, and side rail increased significantly after the training. These questions were related to fall prevention.

## Discussion

Patient safety education is becoming increasingly prevalent in medical education worldwide. In this study, we evaluated the effect of patient safety education on students through

simulation training. Hazardous events or proximity errors related to patient safety can be life-threatening if not promptly and appropriately handled in real clinical situations. However, patient safety education is difficult to acquire in real clinical situations. Therefore, training related to patient safety education is largely provided in the form of lectures. Although the lecture method is effective for transferring knowledge, it is limited in that the acquired knowledge cannot be directly applied to the actual field of practice where various problems may occur [18]. In this regard, simulation training, which is used to simulate medical accidents that occur in actual clinical settings, has been suggested to overcome the limitations of the lecture method, as it creates the possibility of repeated learning and correcting mistakes [10, 19]. Simulation and team-based training are recommended methods to improve patient safety [20]. Therefore, we conducted simulation training for patient safety education.

The results showed significant differences in students' perceptions of patient safety before and after simulation training. We conducted one single training session, and no significant difference was observed in students' attitudes toward patient safety before and after the training; however, the results obtained both before and after the training were confirmed to be above neutral. The results suggest that the training may have helped to form students' basic attitudes toward patient safety. Prior to this simulation training, the students took a course on patient safety and thus had prior knowledge of it. A previous study of medical students reported that the overall attitude scores improved after systematic education, including patient safety education, was provided as part of the regular curriculum [15]. Unlike in this study, educational activities such as lectures and discussions were conducted several times, which may have contributed to the improvement in students' attitudes. This means that improving students' attitudes toward patient safety through one-time education is difficult, and continuous education through a systematic curriculum is necessary.

Self-efficacy in simulation training is the degree of learners' confidence in providing treatment to patients and performing skills in simulated situations [21]. In this study, self-efficacy related to patient safety increased significantly after the simulation training, indicating its efficacy. A previous systematic literature review of simulation-based education reported that simulation training improves students' knowledge, critical thinking, and confidence or satisfaction [22]. Improving students' confidence is an important effect of simulation training, and these aspects are consistent with our results. A previous study that conducted team-based simulation training on patient safety for medical personnel demonstrated an increase in self-efficacy after the training [23]. Another study demonstrated improved self-efficacy among learners after simulation training with a high-fidelity simulator [24]. These results are consistent with those of this study. Further, an increase in students' confidence owing to simulation training is known to improve their performance ability through motivation, as well as their problem-solving and clinical judgment abilities [25]. Therefore, continuous simulation training can help to improve students' self-efficacy related to patient safety and their clinical performance. In other words, future patient safety education should appropriately utilize both theoretical classes and simulation training.

This study had several limitations. First, we evaluated students' subjective responses and did not include objective evaluations of technical aspects. We are preparing a long-term observational study to investigate how they apply to clinical actions. Second, as the simulation training was conducted as part of a medical school's regular curriculum, with five groups alternately engaging in simulation activities every four weeks, this aspect cannot be completely excluded. Third, since the simulation training was part of the regular curriculum, it was impossible to randomly assign students or set a control group. Therefore, the selection of a single group for the study's purpose was inevitable. Fourth, significant changes were found in only 6 of the 30 survey items we used in this study. Only three items in attitude and three items in

self-efficacy showed significant changes, and these items mainly influenced the change in the overall average score. Therefore, the results may be difficult to interpret with only the change in the overall average, and focusing on each significant item is necessary. Fifth, the results may not be generalizable, and since only one medical school was included, the number of samples was small.

## Conclusions

We aimed to evaluate students' perceptions of patient safety by conducting simulation training using a high-fidelity simulator. We focused on falls as one of the most common accidents related to patient safety. The findings demonstrated no significant difference in students' attitudes toward patient safety after the simulation training, but students' self-efficacy was improved when an event related to patient safety occurred.

The result that there was no difference in attitudes toward patient safety suggests that inducing improvement of students' attitudes through one-time training is not sufficient. Therefore, systematic education programs and repetitive simulation training will be required to effectively improve attitudes. In particular, the patient safety training program should be expanded to include high-fidelity simulators that will help simulate actual clinical situations.

We also confirmed the positive effects of simulation training through improved self-efficacy for patient safety. Simulation training must be used in the medical education field continuously to improve patient safety capabilities.

## Acknowledgments

We would like to thank Editage (www.editage.co.kr) for English language editing.

## Author Contributions

**Conceptualization:** Ji Eun Lee, Ji Hye Yu, Su Kyung Lee, Jang Hoon Lee, Hyun Joo Jung.

**Formal analysis:** Ji Eun Lee, Ji Hye Yu.

**Investigation:** Ji Eun Lee, Su Kyung Lee.

**Methodology:** Ji Eun Lee, Ji Hye Yu.

**Supervision:** Ji Eun Lee, Ji Hye Yu, Jang Hoon Lee, Hyun Joo Jung.

**Writing – original draft:** Ji Eun Lee, Hyun Joo Jung.

**Writing – review & editing:** Ji Eun Lee, Ji Hye Yu, Su Kyung Lee, Jang Hoon Lee, Hyun Joo Jung.

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
