## [Decision Letter · Decision Letter 0]

28 Apr 2023

PONE-D-22-35534Comparison of medical students' perceptions of patient safety: Focusing on siulation training using a high-fidelity simulatorPLOS ONE

Dear Dr. Jung,

Thank you for submitting your manuscript to PLOS ONE. After careful consideration, we feel that it has merit but does not fully meet PLOS ONE’s publication criteria as it currently stands. Therefore, we invite you to submit a revised version of the manuscript that addresses the points raised during the review process.

This study is an interesting and important paper on patient safety in medical education. However, in the introduction more background on the simulation used in safety courses should be given.

In the method, there should be more detailed description of the research group and the setting. Both reviewers mentioned that there is more elaboration needed on how you educate the students, what ‘s the content of their training. The scenarios run in the sessions and how they were implemented and what were the main points of debriefing should be explained more clearly. According to Reviewer 1; the questionnaire has no validation. The formation process of the questionnaire details should be given more explicitly.

The results should be clarified and the discuss should be enriched with relevant related studies.

We look forward to receiving your revised manuscript.

Kind regards,

Ipek Gonullu, M.D., Ph.D.

Academic Editor

PLOS ONE

Reviewers' comments:

Reviewer's Responses to Questions

**Comments to the Author**

1. Is the manuscript technically sound, and do the data support the conclusions?

Reviewer #1: Partly

Reviewer #2: Partly

2. Has the statistical analysis been performed appropriately and rigorously? 

Reviewer #1: No

Reviewer #2: I Don't Know

3. Have the authors made all data underlying the findings in their manuscript fully available?

Reviewer #1: Yes

Reviewer #2: Yes

4. Is the manuscript presented in an intelligible fashion and written in standard English?

Reviewer #1: No

Reviewer #2: Yes

5. Review Comments to the Author

Reviewer #1: Respected Authors,

Thank you for considering such a vital topic. However, I do encourage further work on the manuscript.

First, I would like more background on the simulation used in safety courses. We know it enables practice in a safe environment, but I would like you to Focus on the immediate connection with your text.

Not explicit, the introduction is too long when a whole manuscript is considered.

Additionally, there is no detailed description of the research group or the setting; the results are mixed with the methodology.

We need more details on the scenarios run in the sessions and how they were implemented- all eight students a once-at-one scenario. I would not describe debriefing as feedback; it has a broader function- please list which debriefing method was used and what were the main points of debriefing. We have no doubts that a qualified person runs it in medical school—more details on the process. To few details are given on the questionnaires used.

The results are very basic, it is a very simple pre-post comparison. We have no validation of the questionnaire. Please re-think the calculations and how they apply to the clinical actions. There may be more data to work on.

Conclusions are just repetitions of the results.

Therefore, I cannot recommend the text for publication in its current form.

Reviewer #2: Many thanks for the authors to tackle this important subject (patient safety education), please find my comments to improve the manuscript

1. In general an English language editing is required to decrease the word count in certain parts

2. Introduction: no need for too much details on benefit of simulation education

3. Method: how you educate the students , elaborate more on the content of their training , you only mentioned the process

4. Results: a. No 4 and 15 , 19 in table 1 the same data reading noticed in pre and post survey but the p value show highly significant????

b. the table headings is it P value or t value

c. Total in the table is not clear is it the sum , average??? you need to clarify more

d. No need to repeat the result of the table in the written text just concentrate on the main changes

In Discussion it is not clear weather your students have repeated training or it was single training opportunity (2nd paragraph)

6. PLOS authors have the option to publish the peer review history of their article (what does this mean?). If published, this will include your full peer review and any attached files.

Reviewer #1: No

Reviewer #2: No

---

## [Author Response · Author response to Decision Letter 0]

5 Jul 2023

Dear Editor:

We would like to re-submit the manuscript titled, “Comparison of medical students’ perceptions of patient safety: Focusing on simulation training using a high-fidelity simulator”. The manuscript number is PONE-D-22-35534. We thank you and the reviewers for your thoughtful suggestions and insights. The manuscript has benefited from these insightful suggestions. We look forward to working with you and the reviewers to move this manuscript closer to publication in PLOS ONE. The manuscript has been rechecked, and the necessary changes have been made in accordance with the reviewer’s suggestions. The responses to all comments have been prepared and are attached herewith.

Thank you for your consideration. We look forward to hearing from you.

Sincerely,

Hyun Joo Jung

Reviewer 1

1. First, I would like more background on the simulation used in safety courses. We know it enables practice in a safe environment, but I would like you to Focus on the immediate connection with your text.

Response: In Korea, more research is being conducted on patient safety education for nursing students. We have included some information on this and cited the relevant studies. In addition, some sentences have been modified, focusing on the natural connection of the text. (Revised manuscript page 3, line 10-15)

2. Not explicit, the introduction is too long when a whole manuscript is considered.

Response: Thank you for pointing this out. The background of this study was explained in detail so that even those who are not familiar with the subject can easily understand it. However, we agree that the introduction could be shortened, and we have revised it accordingly.

3. Additionally, there is no detailed description of the research group or the setting; the results are mixed with the methodology.

We need more details on the scenarios run in the sessions and how they were implemented- all eight students a once-at-one scenario. I would not describe debriefing as feedback; it has a broader function- please list which debriefing method was used and what were the main points of debriefing. We have no doubts that a qualified person runs it in medical school—more details on the process. To few details are given on the questionnaires used.

The results are very basic, it is a very simple pre-post comparison. We have no validation of the questionnaire. Please re-think the calculations and how they apply to the clinical actions. There may be more data to work on.

Response: We agree with your suggestions. Thus, we added more details on the simulation training program in the form of figures. We hope that these additions clarify how our training program was run. 

In terms of more data, the only available data that we had were the pre-post survey data that investigated students’ perceptions. However, we agree that this can be taken further, and we are preparing a long-term observational study with more medical school students. We will design a study with your insightful suggestions in mind. We have added to list this as a limitation. (Revised manuscript page 3, line 10-15)

4. Conclusions are just repetitions of the results.

Response: We have revised or deleted the repetitive text (Revised manuscript page 7-9). In the conclusion section, we now discuss what we want to emphasize and what the future direction of education should be.

Reviewer 2

1. In general an English language editing is required to decrease the word count in certain parts'

Response: As per your suggestion, we have consulted Editage, an English language editing company, to correct all grammar- and language-related errors and reduce the word count.

2. Introduction: no need for too much details on benefit of simulation education

Response: We agree and have revised the introduction to briefly explain the benefits of simulation education. (Revised manuscript page 3, line 62-70)

3. Method: how you educate the students , elaborate more on the content of their training , you only mentioned the process

Response: We added more details of the simulation training program in the form of figures to provide more details of the training program. (Revised manuscript page 5, line 104-108)

4. Results: a. No 4 and 15 , 19 in table 1 the same data reading noticed in pre and post survey but the p value show highly significant????

b. the table headings is it P value or t value

c. Total in the table is not clear is it the sum , average??? you need to clarify more

d. No need to repeat the result of the table in the written text just concentrate on the main changes

Response: Thank you for pointing out these issues. Our point-by-point responses are as follows:

a & b: We made a mistake in the process of making the table, for which we apologize, and we have now revised it. The previous table did not contain p-values, which may have caused confusion. Therefore, we have now added them to the table. In the previous table, the * mark (p-value<0.05) for No. 4, 15, and 19 was an error. The significant questions were No. 3, 20, 24, 25, 26, and 30. We have revised the table to clarify our results.

c. In the previous table, “Total” referred to the average value of the sub-variable question items. We have revised the terminology used to make this clearer.

d. As per your suggestion, we have removed repetitive information from the text.

(Revised manuscript page 7-9, In Results)

5. In Discussion it is not clear weather your students have repeated training or it was single training opportunity (2nd paragraph)

Response: We conducted one single training session, and we have specified this in the discussion section. (Revised manuscript page 9, In Discussion, 2nd paragraph)

Editor

Response: We have re-checked the formatting requirements and amended the document to adhere to them, including the file name. 

Response: Since we conducted a study on a small number of people, all survey data were written on paper. Therefore, we do not have URLs or DOIs that contain uploaded online data. The original data have been stored in the data storage room of the ACE center at Ajou University Medical School with the approval of Ajou University IRB (Approved No. AJIRB-SBR-SUR-21-624). The storage room is always locked. One can request the data at any time through ‘data access committee’ (the following contact information). If PLOS ONE requests the original data, it can be scanned and sent to you. In addition, since the original data are in Korean, we will translate the questionnaire into English and attach it as a supplement.

[Data Access Committee]

Ji Eun Lee

- Tel. : 82-10-3929-7011

- E-mail : jieun920733@gmail.com

- Hyperlink : Instagram @nueiz.e

Ji Hye Yu

- Tel. : 82-31-219-4092

- E-mail : blue8106@aumc.ac.kr

- Hyperlink : no available hyperlink

Su Kyung Lee

- Tel. : 82-31-219-4463

- E-mail : esukyung71@gmail.com

- Hyperlink : no available hyperlink

Jang Hoon Lee

- Tel. : 82-31-219-5167

- E-mail : neopedlee@aumc.ac.kr

- Hyperlink : no available hyperlink

Hyun Joo Jung

- Tel. : 82-31-219-5160

- E-mail : free1109@ajou.ac.kr

- Hyperlink : no available hyperlink

[Ethics Committee]

Ajou University IRB

- Tel. : 82-31-219-5569

- E-mail : aumc_qa@aumc.ac.kr

- Hyperlink : https://eirb.ajoumc.or.kr/

---

## [Decision Letter · Decision Letter 1]

10 Oct 2023

PONE-D-22-35534R1Comparison of medical students' perceptions of patient safety: Focusing on simulation training using a high-fidelity simulatorPLOS ONE

Dear Dr. Jung,

Thank you for submitting your manuscript to PLOS ONE. After careful consideration, we feel that it has merit but does not fully meet PLOS ONE’s publication criteria as it currently stands. Therefore, we invite you to submit a revised version of the manuscript that addresses the points raised during the review process.

We look forward to receiving your revised manuscript.

Kind regards,

Ipek Gonullu, M.D., Ph.D.

Academic Editor

PLOS ONE

Additional Editor Comments:

Although this is an interesting study on an important topic, it needs more improvement. In the introduction there should be clearer delineation of the need for this work and how this study would contribute to the literature.

In the method, there is more elaboration needed on how you educate the students, what ‘s the content of their training. As Figure 1 and 2 are not enough, the scenarios run in the sessions and how they were implemented and what were the main points of debriefing should be explained more clearly. Besides the questionnaire used for the study has no validation. Simple pre-post comparison results may be assessed/calculated by re-thinking how they apply to the clinical actions.

Reviewers' comments:

Reviewer's Responses to Questions

**Comments to the Author**

1. If the authors have adequately addressed your comments raised in a previous round of review and you feel that this manuscript is now acceptable for publication, you may indicate that here to bypass the “Comments to the Author” section, enter your conflict of interest statement in the “Confidential to Editor” section, and submit your "Accept" recommendation.

Reviewer #3: All comments have been addressed

Reviewer #4: (No Response)

2. Is the manuscript technically sound, and do the data support the conclusions?

Reviewer #3: Yes

Reviewer #4: Partly

3. Has the statistical analysis been performed appropriately and rigorously? 

Reviewer #3: Yes

Reviewer #4: Yes

4. Have the authors made all data underlying the findings in their manuscript fully available?

Reviewer #3: Yes

Reviewer #4: No

5. Is the manuscript presented in an intelligible fashion and written in standard English?

Reviewer #3: Yes

Reviewer #4: Yes

6. Review Comments to the Author

Reviewer #3: They have addressed all the concerned issues, they have shortened the introduction, revised the statistical analysis, I think, it should be accepted

Reviewer #4: This is a Kirkptarick Level 1, pre-post survey design study of 40 medical students at a single medical school during a single pediatric rotation undergoing a specific simulation-based education experience. As such, the findings will be necessarily limited in scope. Since this is a small and new intervention being trialed, it may be better to think of this as a pilot program/QI project. Especially as the authors cannot control for other educational interventions occurring during the 4 weeks of the clerkship that may have affected knowledge, skills, and attitudes beyond their simulation scenarios, which further limits the ability to conclude that their specific intervention is responsible for the results.

Specific comments:

Having read the previous reviewer comments, I feel that the introduction remains too long and is not germane to the study itself. The introduction should focus on simulation safety education literature and how this study would contribute to the literature/answer an unanswered question. Since simulation education like this has been provided for many, as has patient safety education, there needs to be a clearer delineation of the need for this work. Page 4, lines 73-75 are repetitive of a point made in the preceding paragraph; one or the other can be eliminated. Apropos to better delineation of the need for the study, since this is a topic that has been explored before and has already been shown in Korea in nursing students, why should simulation education affect on medical students be different?

Materials and Methods:

Both Figures 1 and 2 do not appear in the copy of the manuscript that I am able to access. Their legends do, as does the entirety of Table 1, so I am unable to state specifically anything regarding their effectiveness/appropriateness. In the Design and Setting, very little demographic data is presented (28 males, 12 females) -- what else is known about the survey/study participants? Ages? Experiences (either work, research, or clinical) that may have affected their answers? More information is needed.

Under Simulation Program Setting (page 5, lines 113-4), the statement is made that debriefing was provided to provide feedback on the learners' knowledge, skills, and attitude. This is not enough information, and does not address adequat5ely previous reviewer concerns. was a specific tool utilized to ensure that all the objectives of the simulation were achieved? did the students switch roles (they were paired up, but it does not say if they had different roles and then did the scenario again after switching roles)?

Results and Disussion:

Of the 30 questions, only 6 had statistically signfiicant changes in responses, 3 in attitude and 3 in self-efficacy. Average score across 24 attitude items did not change. Average score across the 6 self-efficacy items did, which was the predominant driver of the overall average over 30 questions of a statistically significant average score. I find the limited number of items that showed change less convincing of the effectiveness of the education than the authors do, and feel that the conclusions are too broad compared to the limited effects demonstrated in the results section.

7. PLOS authors have the option to publish the peer review history of their article (what does this mean?). If published, this will include your full peer review and any attached files.

Reviewer #3: No

Reviewer #4: No

---

## [Author Response · Author response to Decision Letter 1]

6 Dec 2023

Reviewer 3

1. They have addressed all the concerned issues, they have shortened the introduction, revised the statistical analysis, I think, it should be accepted.

Response: Thank you for your affirmative comment. We found the comments shared by the reviewers from the previous round of review to be valid and valuable, and revised our manuscript accordingly.

Reviewer 4

1. Having read the previous reviewer comments, I feel that the introduction remains too long and is not germane to the study itself. The introduction should focus on simulation safety education literature and how this study would contribute to the literature/answer an unanswered question. Since simulation education like this has been provided for many, as has patient safety education, there needs to be a clearer delineation of the need for this work. Page 4, lines 73-75 are repetitive of a point made in the preceding paragraph; one or the other can be eliminated. Apropos to better delineation of the need for the study, since this is a topic that has been explored before and has already been shown in Korea in nursing students, why should simulation education affect on medical students be different?

Response: Thank you for your insightful comment. In Korea, because the roles of doctors and nurses within hospitals are clearly different, it was believed that there should be differences in education depending on each job group. The duties of doctors and nurses are strictly divided, and doctors have the full and final responsibility for patient safety and management. Therefore, patient safety education for medical students and establishing awareness of patient safety through this education are very important in the medical student curriculum. In addition, since medical schools in Korea have not yet established a school-level curriculum, there is a need to try various methods and determine educational directions. In this regard, the manuscript was slightly modified [Introduction, paragraph 3]. Regarding lines 73–75 on page 4, the manuscript was revised because we agree with the opinion that there is some overlap with the previous paragraph.

2. Both Figures 1 and 2 do not appear in the copy of the manuscript that I am able to access. Their legends do, as does the entirety of Table 1, so I am unable to state specifically anything regarding their effectiveness/appropriateness. In the Design and Setting, very little demographic data is presented (28 males, 12 females) -- what else is known about the survey/study participants? Ages? Experiences (either work, research, or clinical) that may have affected their answers? More information is needed.

Response: Thank you for your helpful comment. Figure 1 illustrates the learning objectives according to the stages of the simulation program and Figure 2 summarizes the process of the simulation program. We believe that there was an error when resubmitting the manuscript; our apologies. We will upload it again.

The students who participated in this study are fifth-year students taking the same curriculum at the medical school. We can provide basic data such as the age of all students who participated in this study and whether they majored in other fields before entering medical school. However, in this study, the survey was conducted anonymously to obtain honest answers from students, so demographic data and the impact of survey answers could not be analyzed.

Specifically, the 40 students who participated in this study were all in their 20s, ranging in age from 22 to 29 years. Among them, one person had a degree in another major (i.e., computer engineering) before entering medical school. Additionally, these students had no other research experience in the field of basic medicine or sub-internship experience before participating in this study.

3. Under Simulation Program Setting (page 5, lines 113-4), the statement is made that debriefing was provided to provide feedback on the learners' knowledge, skills, and attitude. This is not enough information, and does not address adequat5ely previous reviewer concerns. was a specific tool utilized to ensure that all the objectives of the simulation were achieved? did the students switch roles (they were paired up, but it does not say if they had different roles and then did the scenario again after switching roles)?

Response: Thank you for your valuable comment. We conducted debriefing using the “PEARLS healthcare debriefing tool.” Therefore, we did not proceed with the simulation program by switching roles after the debriefing session. We simply surveyed the students about how they felt before and after the simulation training. However, we think that it is necessary to conduct a simulation again after the debriefing to determine how the students’ perceptions level has changed. Therefore, we hope that future research focusing on this topic can be conducted.

4. Of the 30 questions, only 6 had statistically signfiicant changes in responses, 3 in attitude and 3 in self-efficacy. Average score across 24 attitude items did not change. Average score across the 6 self-efficacy items did, which was the predominant driver of the overall average over 30 questions of a statistically significant average score. I find the limited number of items that showed change less convincing of the effectiveness of the education than the authors do, and feel that the conclusions are too broad compared to the limited effects demonstrated in the results section.

Response: Thank you for your insightful comment. We agree with some of your perspectives. Therefore, we have revised the manuscript by mentioning the aspects you have pointed out in the paragraph where we present our limitations [Discussion, paragraph 4]. The number of items showing significant changes is small, which may seem unconvincing. However, it can be seen that just one training session can sufficiently increase students’ self-efficacy. Attitudes, meanwhile, seem difficult to change with one-time training. Therefore, we believe that repetitive training is necessary, and as mentioned in the paper, we are preparing a study to examine the effects of long-term training.

---

## [Decision Letter · Decision Letter 2]

29 Jan 2024

PONE-D-22-35534R2Comparison of medical students' perceptions of patient safety: Focusing on simulation training using a high-fidelity simulatorPLOS ONE

Dear Dr. Jung,

Thank you for submitting your manuscript to PLOS ONE. After careful consideration, we feel that it has merit but does not fully meet PLOS ONE’s publication criteria as it currently stands. Therefore, we invite you to submit a revised version of the manuscript that addresses the points raised during the review process.

I thank the authors for addressing the reviewers’ comments. However, it needs more improvement about the necessity of this study in terms of patient safety and simulation training together.

The simulation program was not described clearly (development, the content of their training, scenarios, debriefing ,…)

We look forward to receiving your revised manuscript.

Kind regards,

Ipek Gonullu, M.D., Ph.D.

Academic Editor

PLOS ONE

Reviewers' comments:

Reviewer's Responses to Questions

**Comments to the Author**

1. If the authors have adequately addressed your comments raised in a previous round of review and you feel that this manuscript is now acceptable for publication, you may indicate that here to bypass the “Comments to the Author” section, enter your conflict of interest statement in the “Confidential to Editor” section, and submit your "Accept" recommendation.

Reviewer #5: (No Response)

Reviewer #6: All comments have been addressed

2. Is the manuscript technically sound, and do the data support the conclusions?

Reviewer #5: Partly

Reviewer #6: Yes

3. Has the statistical analysis been performed appropriately and rigorously? 

Reviewer #5: Yes

Reviewer #6: Yes

4. Have the authors made all data underlying the findings in their manuscript fully available?

Reviewer #5: Yes

Reviewer #6: Yes

5. Is the manuscript presented in an intelligible fashion and written in standard English?

Reviewer #5: Yes

Reviewer #6: Yes

6. Review Comments to the Author

Reviewer #5: The manuscript seems to have been revised a lot, according to previous reviewer comments. However, in my opinion, to reflect their comments, further revision would be necessary. First, the necessity of this study should be mentioned in terms of patient safety and simulating training together. In the introduction, the authors seem to be dealing with the necessity of this study in terms of patient safety and simulating training separately. This discrepancy weakens the necessity of this study. 

Second, Figure 2 does not reflect the process of the simulation program’. It looks like the process of this study. Under the title'simulation program setting’, the authors do not carefully use the terms'simulation program' and'study' differently, which confuses readers about the study between the traning programs. Fig. 2 should be included under the title “Design and Setting."

Third, along with the Fig. 2 problem, the study design should be more detailed to determine, when the survey was taken, whether there was no difference in simulation experience due to the rotation experience (since for most medical schools in Korea, students’ previous experiences are different due to the rotation assignment). Furthermore, authors should describe the development of simulation programs, whether they adopt previous simulation programs based upon previous studies, etc. It is hard to compare the efficacy of the program if there are any reference points.

Reviewer #6: I thought the previous version was acceptable and the current additional revision is acceptable. At some point, the data need to be made available to the readers for scrutiny.

7. PLOS authors have the option to publish the peer review history of their article (what does this mean?). If published, this will include your full peer review and any attached files.

Reviewer #5: No

Reviewer #6: No

---

## [Author Response · Author response to Decision Letter 2]

4 Mar 2024

Reviewer 5

1. The manuscript seems to have been revised a lot, according to previous reviewer comments. However, in my opinion, to reflect their comments, further revision would be necessary. First, the necessity of this study should be mentioned in terms of patient safety and simulating training together. In the introduction, the authors seem to be dealing with the necessity of this study in terms of patient safety and simulating training separately. This discrepancy weakens the necessity of this study.

Response: Thank you for your valuable comment. We reviewed the introduction again. In response to your comment, we have added brief information about the link between patient safety and simulation education. [Introduction, paragraph 4]

2. Second, Figure 2 does not reflect the process of the simulation program’. It looks like the process of this study. Under the title'simulation program setting’, the authors do not carefully use the terms'simulation program' and'study' differently, which confuses readers about the study between the traning programs. Fig. 2 should be included under the title “Design and Setting."

Response: Thank you for your insightful comment. We reviewed Fig. 2 and agree with your opinion that it was closer to a study design than a simulation program setting. Hence, we inserted the figure into the “Design and Setting” section of our manuscript. [Materials and Methods-Design and setting, paragraph 2]

3. Third, along with the Fig. 2 problem, the study design should be more detailed to determine, when the survey was taken, whether there was no difference in simulation experience due to the rotation experience (since for most medical schools in Korea, students’ previous experiences are different due to the rotation assignment). Furthermore, authors should describe the development of simulation programs, whether they adopt previous simulation programs based upon previous studies, etc. It is hard to compare the efficacy of the program if there are any reference points.

Response: Thank you for your helpful comment. The students who participated in this study are all studying the same curriculum in one medical school. Therefore, their experience before participating in pediatric clinical practice is also the same. Additionally, since simulation practice is not performed in other clinical departments that rotate at the same time, there is no difference. Furthermore, among various topics related to patient safety, which are important emerging issues, falls were selected as one of the most common accidents in pediatric departments. [Materials and Methods-Simulation program setting, paragraph 1] Although many references were reviewed to set up the simulation program, the program’s contents were developed independently, regardless of these references.

Reviewer 6

1. I thought the previous version was acceptable and the current additional revision is acceptable. At some point, the data need to be made available to the readers for scrutiny.

Response: Thank you for your affirmative comment. We revised the manuscript based on the comments and suggestions of the reviewers. We hope our manuscript improves further such that it is suitable for publication in your esteemed journal.

---

## [Decision Letter · Decision Letter 3]

21 May 2024

Comparison of medical students' perceptions of patient safety: Focusing on simulation training using a high-fidelity simulator

PONE-D-22-35534R3

Dear Dr. Jung,

We’re pleased to inform you that your manuscript has been judged scientifically suitable for publication and will be formally accepted for publication once it meets all outstanding technical requirements.

Kind regards,

Ipek Gonullu, M.D., Ph.D.

Academic Editor

PLOS ONE

Additional Editor Comments (optional):

Reviewers' comments:

Reviewer's Responses to Questions

**Comments to the Author**

1. If the authors have adequately addressed your comments raised in a previous round of review and you feel that this manuscript is now acceptable for publication, you may indicate that here to bypass the “Comments to the Author” section, enter your conflict of interest statement in the “Confidential to Editor” section, and submit your "Accept" recommendation.

Reviewer #4: All comments have been addressed

Reviewer #5: All comments have been addressed

2. Is the manuscript technically sound, and do the data support the conclusions?

Reviewer #4: Partly

Reviewer #5: Yes

3. Has the statistical analysis been performed appropriately and rigorously? 

Reviewer #4: Yes

Reviewer #5: Yes

4. Have the authors made all data underlying the findings in their manuscript fully available?

Reviewer #4: No

Reviewer #5: Yes

5. Is the manuscript presented in an intelligible fashion and written in standard English?

Reviewer #4: Yes

Reviewer #5: Yes

6. Review Comments to the Author

Reviewer #4: The authors have addressed my previously stated concerns. Lines 42 and 43 in the introduction (page 2 of the PDF) are duplicative from the paragraph before and can be deleted. The introduction still reads too long and could be more concise to get to the study itself faster, but these are minor quibbles.

Reviewer #5: The author revised the manuscript according to the comments sufficiently. There is no further suggestions.

7. PLOS authors have the option to publish the peer review history of their article (what does this mean?). If published, this will include your full peer review and any attached files.

Reviewer #4: No

Reviewer #5: No

---

## [Editor Report · Acceptance letter]

9 Jul 2024

PONE-D-22-35534R3 

PLOS ONE

Dear Dr. Jung, 

I'm pleased to inform you that your manuscript has been deemed suitable for publication in PLOS ONE. Congratulations! Your manuscript is now being handed over to our production team.

Kind regards, 

on behalf of

Associate Professor Ipek Gonullu 

Academic Editor

PLOS ONE